# Interactive Visualisation of Sustainability Indicators for Water, Energy and Food Innovations

**Paula J. Forbes** [1,*] , **Ruth E. Falconer** [1] , **Daniel Gilmour** [2] **and Nikolay Panayotov** [1]

1   School of Design & Informatics, Abertay University, Dundee DD1 1HG, UK; r.falconer@abertay.ac.uk (R.E.F.); n.panayotov@abertay.ac.uk (N.P.)
2   School of Applied Sciences, Abertay University, Dundee DD1 1HG, UK; d.gilmour@abertay.ac.uk
*   Correspondence: p.forbes@abertay.ac.uk

**Abstract:** The Water-Energy-Food (WEF) nexus describes the synergies and trade-offs between water, energy and food. Despite the significant attention that the WEF nexus has received in recent years, challenges remain, primarily related to gaps in integrated data, information and knowledge related to the most critical inter-linkages and their dynamics. These WEF nexus complexities and uncertainty make decision-making and future forecasting extremely difficult. Policy makers and other stakeholders are currently faced with the task of understanding longer term environmental impacts and tJhe benefits and limitations of innovations that could be potentially beneficial, such as Anaerobic Digestion as a waste solution or insect protein production. This paper describes an approach to support decision making for local-level innovations within the WEF nexus by creating a set of sustainability indicators and an accompanying interactive visualisation. The indicators were derived from stakeholder consultation processes and workshops, and they were selected to include a much broader assessment than just financial aspects when considering the viability of such innovations. By taking this bottom-up approach and placing stakeholders at the heart of the project, we produced a visualisation tool to support sustainable decision making when considering the implementation of WEF innovations. Considering other, often overlooked factors and giving greater priority to these deepens knowledge and the recognition of influential issues that in conventional processes may be overlooked. This visualisation tool is designed to support decision makers to engage in a exploration of the different interlinkages, and to be the basis of stakeholder dialogue around sustainability. The visualisation tool developed was designed to be easily modifiable in order to be updated with new insights and to include other future innovations.

**Keywords:** sustainability indicators; WEF nexus; decision support; sustainability innovations; complexity; AD; interactive visualisation; SDGs



## 1. Introduction

With depleting natural resources; pollution of our air, land and seas; and increasing global warming, making good decisions to minimise environmental impact is essential. Many global challenges, despite being interconnected are often addressed individually, and this can lead to problems in other areas whilst trying to reduce a specific issue. The Water-Energy-Food (WEF) nexus was a term coined by the World Economic Forum in 2011 and it describes the synergies and trade-offs between water, energy and food. However, WEF nexus complexities and uncertainty make decision-making and future forecasting extremely difficult. Policy makers and other stakeholders are currently faced with the task of visualising longer term environmental impacts and the benefits and limitations of innovations that could be potentially beneficial, such as Anaerobic Digestion as a waste solution or insect protein production.

There has been increasing interest over the past decade or so of how to promote and govern a transition towards sustainability with regards to food and energy production and

consumption and the effective management of valuable water resources. Informed and integrated decision making is a priority for policy makers at the national and international level. The interconnected nature of the processes involved requires the capacity to understand complex issues across multiple disciplines and sectors. The WEF nexus has become a powerful frame through which the interactions and interdependencies of the three systems (Water, Energy & Food) can be examined, and the aim of much of the nexus research is to promote understanding and to develop tools that can assess and communicate these interdependencies [1]. Problem solving for sustainability in the framework of the WEF nexus is likely to become even more challenging due to the impacts of population growth, climate change, increasing urbanisation, global dietary changes and the interconnected nature of these issues [2] and the unprecedented speed of these global changes. Nexus oriented resource management is particularly important for us to achieve the UN Sustainable Development Goals (SDGs) [3].

The Sustainability indicators and visualisation tools described in this paper were developed as part of the Stepping Up project http://steppingupnexus.org.uk (accessed on 5 May 2020) (See Supplementary Materials). The other tools developed were an Agent Based Model (for AD) that was combined with Multi Criteria Decision Analysis (MCDA) [4] and also a Serious Game prototype (unpublished). How the decision support tool for county-scale AD was developed is described in a further project publication [5] highlighting how the tool can support exploration of the social, environmental and economic impacts of different AD strategies and decisions. The Stepping Up project undertook research to understand the processes, implications, and challenges of scaling up nexus-innovations to achieve transformational change at different scales. Stepping Up focused on three existing niche innovations that may support the transition towards sustainability: anaerobic digestion, insect protein for feed for animals or food for people and the redistribution of surplus food. These innovations exist within established systems in various contexts and at different scales. They are also not purely technological innovations, although each includes some aspect of novel technological application or development. Some have been in existence for some time, but they are not currently mainstream (in the UK at least). The innovations are linked by the common theme of waste, which can be considered both as a burden and, increasingly, as a resource. In the context of increasing resource scarcity and rising greenhouse gas emissions caused by our rapid population growth, the chosen innovations have the potential to offer insights into the development of new resources across several applications. Building case studies around those innovations will offer us interesting insights into technical, systemic, production driven and consumer driven practices of innovations at different levels of maturity. They represent potentially viable seeds of change, which may or may not provide opportunities for existing systems to adapt to the pressures of societal and environmental challenges. Thus, they have the potential to help transform practices within production and consumption with benefits that span the WEF nexus.

The complexities and uncertainty of the WEF nexus are highlighted by recent research [6,7], and barriers exist to decision making in the UK WEF nexus [8]. The WEF nexus comprises many intertwined social, economic and engineering considerations that cut across the highly heterogenous landscapes of Food, Energy & Water, and there is a growing need to better understand the trade-offs associated with their future management. [9]. A good definition of how this approach is supporting resource management is provided by 3], who states, "The nexus approach is evolving into an integrative concept which bridges sectors and considers interrelated resources in an unbiased way to achieve sustainable resource management." The goal of any WEF nexus assessment is to inform nexus-related responses in terms of strategies, policy measures, planning and institutional set-up or interventions [10]. Adequate stakeholder engagement at all stages of the nexus assessment is a key condition to ensure high quality assessment and response. Including relevant stakeholders in the development of these decision support tools increases their effectiveness and should also encourage trust in their ability to convey the information to

the end users. The vital role played by stakeholders in this research project is discussed in a recent publication by our Stepping Up project [1,11]. Different types of stakeholders will have differing requirements and different levels of understanding about the data presented to them.

The use of Future Scenarios to portray the complexity and multidimensionality of the WEF nexus approach to sustainability innovations has been described in another Stepping Up publication [12]. This paper explains how the sustainability transformations were explored with stakeholders, and how cross-cutting themes such as social issues along with the usual climatic and technological ones were discussed in the context of three of these different possible futures. How Anaerobic Digestion (AD) is used as an innovation example to understand the potential multi-sectoral benefits across the WEF nexus and also the factors influencing the technology uptake are described in [13]. This paper also highlights how a novel mixed-method approach that integrates stakeholder knowledge from multiple disciplines offers significant value, providing a deeper understanding of the enablers and barriers for scaling up a specific innovation. Current feasibility studies for innovations, such as AD, often focus solely on financial aspects over the short to medium term and do not generally consider the wider implications to the Water Energy Food nexus, nor do they usually consider projections further into the future, where possible disruptions to the current status quo may have happened; for example, it is unlikely that anyone modelled the impacts of the ongoing Covid-19 pandemic on food security, energy use or what the impact of this will be in the future. Considering nexus approaches is not an easy task and requires a move from the purely conceptual to the more practical.

Given the immense complexity of the issues and the level of understanding required to make good decisions, there has been significant recent academic interest in the development of Decision Support Tools (DSTs) and their use by policy makers for decreasing the environmental impact of our resource use [14], supporting climate change [15] and managing complexity [16]. Examples of these tools include: MuSIASEM [17,18], WEAP [19] (for water use) and Nexus Tool 2.0 [20], and the capabilities and limitations of these and other tools are discussed by [21] who concluded that gaps exist in the nexus approach, with the toughest limitation in modelling as the nexus approach has extensive data requirements (which are not always available). The difference between these tools and our visualisation tool is that the above examples give specific outputs based on simulations of data, whereas our tool is solely aiming to give the user an indication of the complexities involved in the decision-making process. The CLEWS framework (Climate, Land-Use, Energy Water) illustrates the synergies and trade-offs within the CLEW areas for decision making related to achieving development goals. These tools address WEF nexus issues at varying scales, but it is important to remember that decisions made at small scale (e.g., farm or local level) have an impact at larger scales (Regional, National and even Global). Decision support tools are often based on purely quantitative methods and tend to have a very siloed approach to the calculation or simulations produced. The context specific nature of most decision processes adds to the already complex challenge of deciding on the best use of valuable resources.

The difficulties involved in WEF nexus decision support, such as the disparate and dynamic datasets available (usually for a specific discipline) and the difficulty in designing strategies that are robust under various future scenarios, are discussed by [22]. Methods to support exploratory decision-making under conditions of deep uncertainty are discussed by some of the authors of this paper in the following publication [4].

A recent series of papers [23] concluded that applying a nexus approach is vital for the sustainable use of environmental resources and this will be instrumental in achieving the UN Sustainable Development Goals (SDGs). No studies have, as of yet, explicitly quantified the contribution of nexus approaches in progress towards the SDGs [24]. The approach of linking our indicators to the Sustainable Development Goals was chosen for our visualisation tool as these are well-known globally and can also be represented very well visually using their colourful icons. A similar approach of linking the WEF

nexus and SDGs was taken by [25] to develop a monitoring tool to track the impact of innovations in Mediterranean countries. Challenges relating to the implementation and monitoring of the targets of the SDGs from a water perspective are discussed in [26], where it is recognised that implementing SDGs is a societal process of development, and that there is a need to link how SDGs relate to public benefits and to better communicate this to the broader public.

There are many types of Decision Support Tool (DST), many of which focus on the financial viability aspects of implementing an innovation such as AD, with Return on Investment (ROI) being the main deciding factor. Our stakeholders stressed that financial viability was, of course, of paramount importance. Estimates of ROI can be achieved with relative ease by other tools (although changing financial incentives makes predicting longer-term viability more difficult). The issue of taking a broader, more informed approach was summed-up very well by one of our stakeholders (an advisor for Zero Waste Scotland) when discussing the considerations required for energy production from waste via Anaerobic Digestion:

> *"There are lots of things to consider, it's not as simple as we'll take the food waste and make lots of electricity and have a lovely product at the end of it—it's just not that simple. It's trying to get people to realise that, but in a way that's not being disrespectful of their ideas. I'm not saying it's not going to work, but what I'm saying is please think of all these things because otherwise you are going to be left with a BIG problem at the end of it which will impact you, your stakeholders and investors. All I'm saying is look at it as broadly as you can."*

We, therefore, decided to focus more on the sustainability aspects of the decision around the viability of innovations, as this aligned with the focus of the Stepping Up project. We decided to link our indicators to the globally recognised SDGs in order to give end-users some indication of the larger benefits of the specific innovations that we focussed on. By considering these innovations under three different potential future scenarios, which are described in full in [12], we also include some "food for thought," i.e., considerations of how things could be in varying futures, and this should promote debate and discussion around the decision to be made.

## 2. Materials and Methods

The production of the Sustainability Indicators and the subsequent interactive Visualisation were a result of pulling together a variety of work done by the different Stepping Up project partners, which has been documented in the following publications [1,4,5,12,13]. The initial focus was on collecting relevant and contextual qualitative data from our key stakeholders. This was done by a combination of interviews, Focus Groups and workshops. A large workshop with many different stakeholders was carried out in London to investigate possible future scenarios across the WEF nexus [13], and a smaller workshop was held later in the project to validate some of the indicators that we thought were important. At least 15 interviews were made with different stakeholders, these included:

- Local Council Authorities (Scotland & England);
- AD plant operators, entrepreneurs and experts (at various scales);
- National organisations such as Zero Waste Scotland;
- Food Redistribution organisations (ReFood);
- Innovators producing Insects for Protein.

Interviews were first transcribed, and then analysed and coded in NVivo using grounded theory to identify relevant themes. These themes, along with the themes emerging from the workshops, then informed the Sustainability Indicators and also the Personas for our end-users and their possible future interaction with the DST that we were developing. The Indicators we decided upon after the thematic analysis were then compared with other sustainability indicators from relevant academic publications; we compiled an initial SI table for potential inclusion in the visualisation tool that was then verified with

key stakeholders in a workshop and refined after group discussion. We have categorised these using the three broad pillars of Environment, Social and Economic factors.

## 3. Results

From these interviews with relevant stakeholders, themes emerged highlighting important issues that our indicators should address. To allow us to develop an understanding of the different types of stakeholder's needs, several Personas and their corresponding User Stories were then developed—each designed to be indicative of a different potential end-user.

In writing our user stories, we also considered three different future scenarios that were initially proposed by the project and matched a user story to each of these Scenarios. As the future is uncertain, scenario approaches are designed to understand the societal, climatic, technological and economic implications of these possible futures [1]. The three future scenarios developed by the project for the purpose of exploring with stakeholders, how AD/insect protein production/food redistribution could be scaled up. The visualisation tool does not intend to nudge people in any direction towards any of the three, but to offer users a way of exploring how the three scenarios could impact the innovations and SI's included in the tool. The three future scenarios devised in [13] are shown in the Table 1 below.

**Table 1.** The three possible future scenarios considered.

| Share & Connect | Create & Cope | Big & Smart |
| --- | --- | --- |
| Decentralised digital society with high levels of connection between producers, consumers and the environment. | A society troubled by climate change, but with vibrant innovation in services systems catering for most needs. | A highly centralised society where big infrastructure supplies for basic needs, regulated for transparency and efficiency. |

Figure 1 below shows an abbreviated example of a Persona (Local authority Policy maker) showing goals, motivations and frustrations. The green bars on the top right give an indication of the importance of each category (WEF) to the person's role. We also developed several User Stories to highlight the requirements of a range of end users who could benefit from using the Indicators.

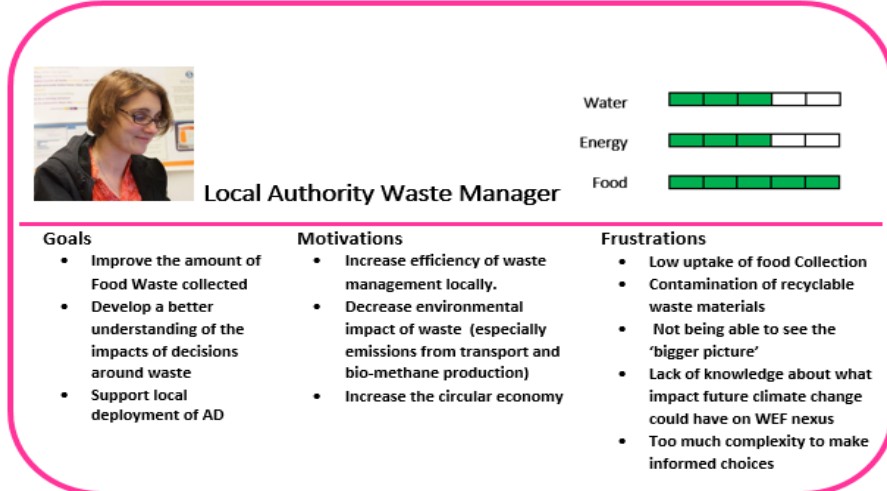

**Figure 1.** Example of a Persona developed for the Stepping Up project Decision Support Tool.

In depth thematic analysis of the interviews and workshops with the various stakeholders was carried out and themes were identified (using NVivo software). This helped to

identify interconnections and gave us a much deeper understanding of the issues around the implementation of the different innovations and the Sustainability Indicators that could be used to support decision making. Table 2 below shows the Sustainability Indicators chosen, the different categories they were placed in, if they were a driver or a barrier to the chosen innovations and if the desired benefit would be an increase or decrease in the indicator measurement.

**Table 2.** Sustainability Indicators.

| Economic | Driver/Barrier | Increase/Decrease |
|---|---|---|
| Transport Costs | Barrier | − |
| Revenue Biogas Generated | Driver | + |
| Gate Fee Costs Waste | Driver | + |
| Running Costs | Barrier | − |
| Build Costs | Barrier | − |
| Revenue from Incentives | Driver | + |
| Revenue Alternative Protein | Driver | + |
| Energy Grid Infrastructure | Driver | + |
| By-product disposal costs | Barrier | − |
| **Environmental** | | |
| Carbon Footprint (reduction) | Driver | − |
| Water consumption (reduction) | Driver | − |
| Water Quality (improvement) | Driver | + |
| NPK fertiliser use (reduction) | Driver | − |
| Energy consumption (reduction) | Driver | − |
| Air Quality (improvement) | Driver | + |
| CO2 production from transport | Barrier | − |
| Soil Quality (improvement) | Driver | + |
| Land Take (amount land required) | Driver | − |
| **Social** | | |
| Resource Redistribution | Driver | + |
| Quality of Life | Driver | + |
| Job Creation | Driver | + |
| Knowledge Sharing | Driver | + |
| Access to relevant Skills | Driver | + |
| Access to relevant Technology | Driver | + |
| Education | Barrier | + |
| Visual Disturbance | Driver | − |
| Social Acceptance | Driver/Barrier | + |
| **Socio-Economic** | | |
| Energy Security | Driver | + |
| Regional Development | Driver | + |
| **Socio-Environmental** | | |
| Food-waste availability | Driver | +/− |
| Waste Regulations | Driver | + |

*Developing the Decision Support Tool*

In creating our Interactive Visualisation, we established which of the SDGs were relevant to each of our selected Sustainability Indicators (SIs), and how they were interconnected with other issues and the innovations themselves. To do this, a card sort exercise was carried out to support the task. The UN SDGs were chosen as they are familiar to people and most countries have signed an agreement to strive towards reaching these goals by 2030. Each of the SIs and SDGs were written on separate Post-It notes, and each individual SI was then selected and placed at the centre of a board and all related SIs and SDGs placed around the central SI.

Figure 2 shows an example of this process, with Bioenergy use as the central SI and the relevant connections made. The connections had been identified by qualitative analysis

of the focus group and interviews with stakeholders and were validated during a later workshop with stakeholders. The resulting collection of Post-It notes were photographed to capture these connections, and then also examined at by a second researcher involved in the project to check if anything had been missed and to ensure that there was a consensus with the choices made. These were then recreated using Miro to give better graphical results than the original Post-It notes. A large spreadsheet was then created that captured all the relevant SIs along with each of the SDGs applicable to it and notes made that related the SI and how it may relate to the different future scenarios. In addition, the table included details of whether the indicator was a driver or a barrier to innovation, and if a beneficial result was for there to be an increase or decrease in the indicator. The scale of the indicator (i.e., local, regional, national or global) was also added as well as which sustainability pillar (Environmental/Economic/Social) the indicator related to.

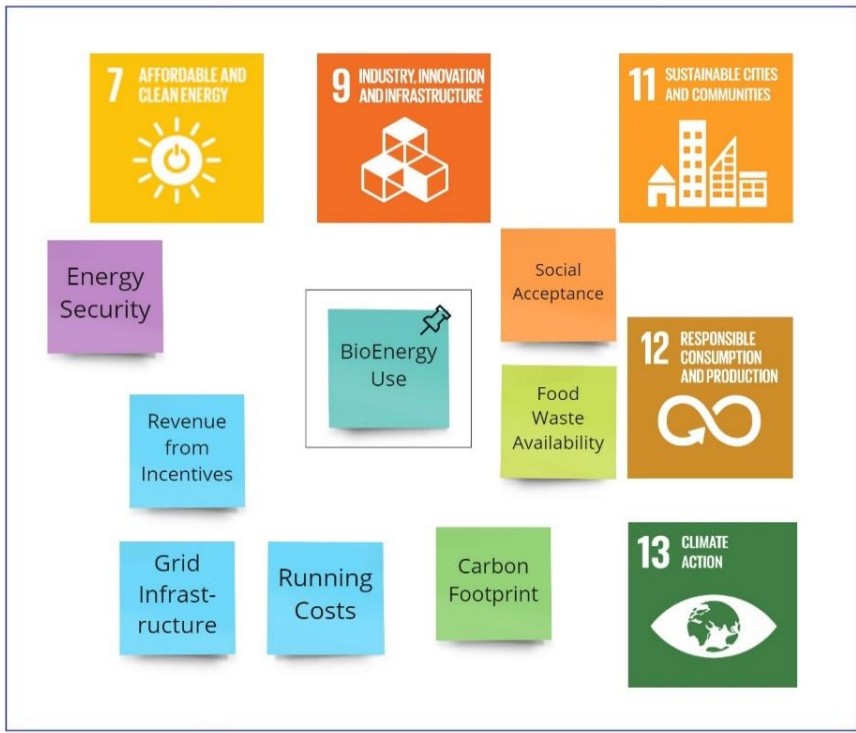

**Figure 2.** Card sort image for achieving connections with other SIs and the SDGs relating to Bioenergy use from Anaerobic Digestion.

The next and final step was to develop the Visualisation in a way that captures the various sustainability indicators from the many identified points of view in a cohesive and visually accessible way, which allows for inquisitive exploration. The finished interactive tool can be found at: https://nikpanayotov.github.io/steppingup-indicators-catalogue/ (accessed on 5 May 2020).

The tool was developed in the form of an online web application. It is powered by HTML5 Open Web Platform technologies, including the open-source data visualisation library D3.js (https://d3js.org/) (accessed on 7 August 2020). The web app can run entirely on the client side, which allows for minimal server setup and maintenance. The code for the web app is also open source and can be found at: https://github.com/NikPanayotov/steppingup-indicators-catalogue (accessed on 5 May 2020).

The web app visualisation tool (DST) developed attempts to display all the relevant indicators in a single view to showcase the scope and diversity of lenses. The visual design takes inspiration from sunburst visualisations, which highlight parts that make up a whole on several levels. The indicators are represented by coloured blocks "bursting" from the centre (shown on the outside of the circle, Figure 3 above). The different indicators

were colour coded to enable better visual clarity, with Economic indicators blue, Social orange, Environmental green, Socio-Economic purple, Socio-Environmental khaki green and where the issue affected all three pillars grey. The user can hover with their mouse over a block to display their name. In the centre of the tool, the user can select one of the three innovations: Insect Protein, Anaerobic Digestion and Food Redistribution. Each icon filters the indicators that come out of the centre according to the selected innovation. If no icon is selected, then all indicators across all innovations are displayed. The second layer can further highlight separate groups of related indicators by categories: land, food waste, bioenergy, economy, education and climate. The circular design emphasises the connected nature of the indicators and categories by their proximity and concentrated, closed form.

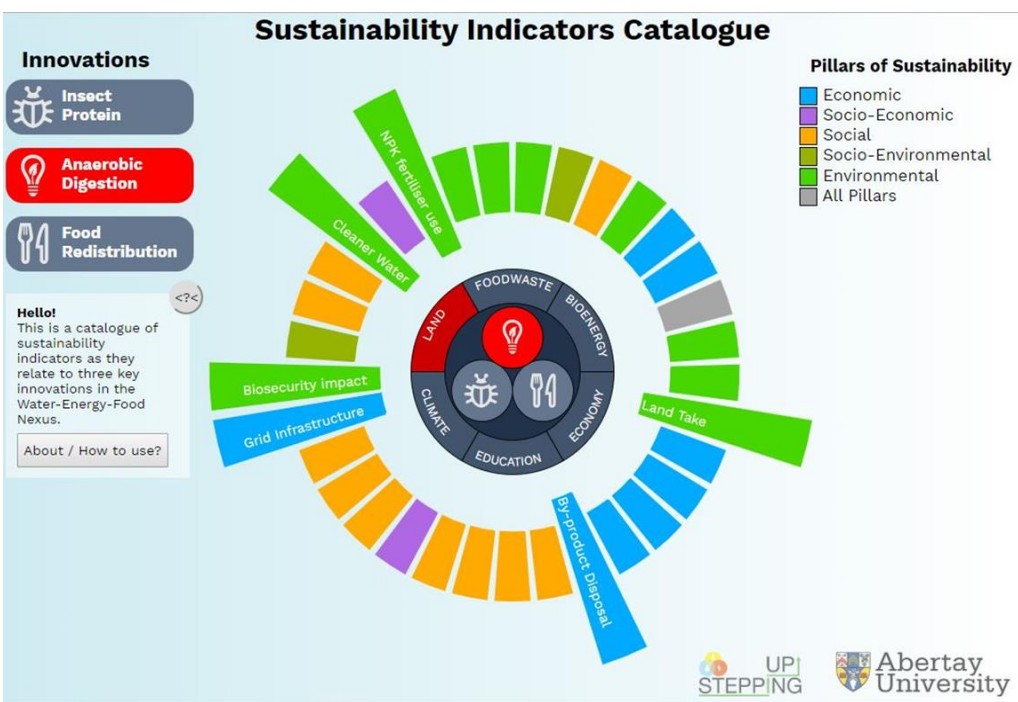

**Figure 3.** Screen shot of the Stepping Up DST showing land-based issues relating to Anaerobic Digestion.

Clicking on a coloured block, representing an individual sustainability indicator, opens up a pop-up frame with more specific information (Figures 4 and 5 below). This screen includes:

- a more detailed definition of the indicator.
- whether the indicator is a driver/barrier and if the aim would be to increase/decrease the indicator.
- a description of the indicator's behaviour or interpretation under the different future scenarios.
- the indicator's relation to the UN Sustainable Development Goals.

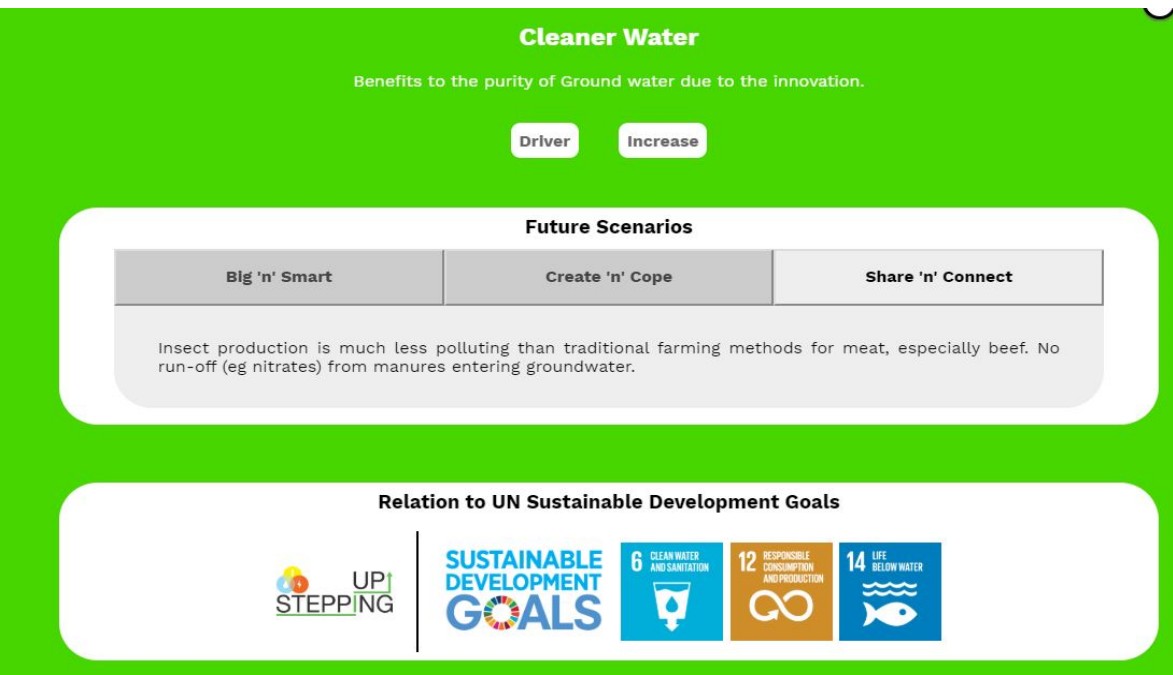

**Figure 4.** Example of Single Sustainability Indicator (SI) display for Future Scenarios & Relationship to SDGs—Cleaner Water.

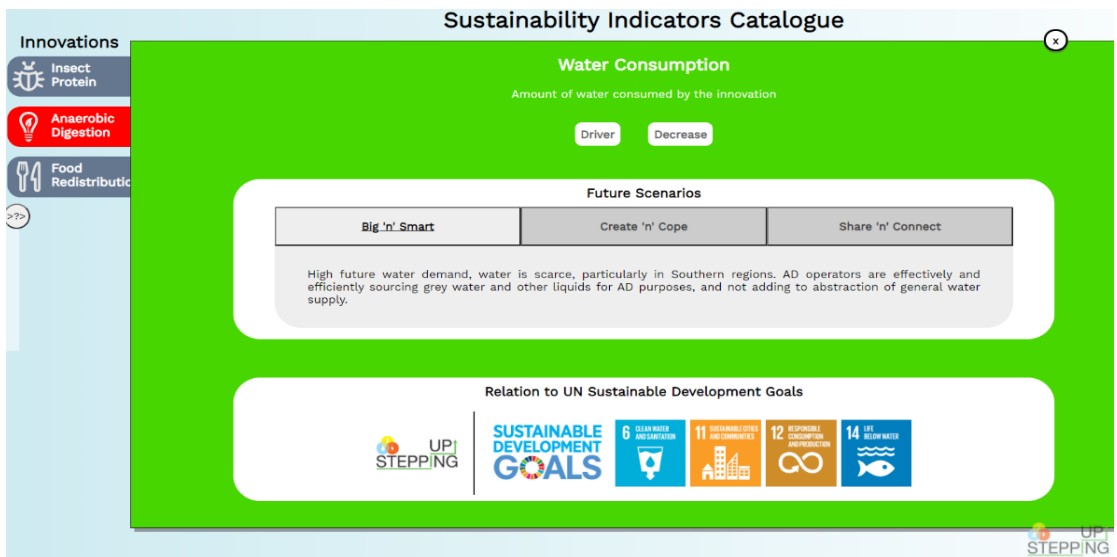

**Figure 5.** Example of Single Sustainability Indicator (SI) display for Future Scenarios & Relationship to SDGs—Water Consumption.

## 4. Conclusions

Despite the significant attention the WEF nexus has received in recent years, challenges remain, primarily related to gaps in integrated data, information and knowledge related to the most critical inter-linkages and their dynamics [24]. Ref. [24] also states that there is a lack of systematic tools that can address all the synergies and trade-offs involved in the nexus. Policy makers and planners usually operate in silos; this is exemplified by the fact that in most countries, different government ministries handle agriculture, energy and health. It is difficult for them to identify and understand which interactions are the most important to address and to evidence which policies help or hinder progress towards the SDGs [27,28].

The vagueness and ambiguity currently associated with the WEF nexus makes it challenging to make sense of the complexity in a way that enables appropriate action [29]. The development of our sustainability indicators and the associated Decision Support Toolkit (DST) attempts to address this complexity; it was not intended to provide any definite "answer" to which innovation may benefit sustainability goals in the future, but to provide a focus for discussion for stakeholders to consider the possibly conflicting criteria across the WEF nexus and under different contexts. It is hoped that the design of our Visualisation tool can communicate the integrated nature of the sustainability indicators and promote a holistic approach to decision making, which avoids thinking in silos. Its single-screen overview presents the full scope of the indicators and their relation to the pillars of sustainability. The interactive design invites inquiry and encourages exploration of the indicators on several levels and from different points of view. The included examples of three innovations present how different indicators might be useful under different application areas. The innovation process is likely to be stifled by solely prioritising financial aspects. Although many stakeholders conveyed the fact that ultimately financial aspects are a deciding factor and tend to drive decisions, they tend to stifle the innovation process when this is the main lens looked through. This is highlighted by the following quote from a stakeholder:

> "Just now the decision (on AD) is made on knowing it's going to cost you an extra £3 million capital and then £1.5 million per year for the food waste collection. You have to decide if AD is a route that you want to go down and match this up against any other options you have. Seeing the environmental and social benefits of the possible choices would really help our understanding of what the best options would be for the future."

It is hoped that by considering other, often overlooked factors, and by giving greater priority to these, that deeper knowledge and the recognition of influential issues that in conventional processes may have been overlooked may come to light. It is evident that this tool is more suitable for some stakeholders than others, and we understand that the Agent Based Modelling for Multi Criteria Decision Analysis (ABM MCDA) tool that was also developed for this project [5] is likely to appeal in circumstances where a more data-driven approach is required. This visualisation tool, although not providing any specific "answers," should allow for a more playful exploration of the different interlinkages, and to be the seed of conversations around sustainability and forge a shared understanding. The visualisation tool developed was designed to be easily modifiable in order to be updated with new insights and to include other innovations. There may be future opportunities to refine the design and to re-iterate it with various stakeholders; the tool can be used to explore the innovations, the indicators and the possible future scenarios in a playful and collaborative manner, and to be the desired conversation starter that it was intended to be.

**Supplementary Materials:** A series of 4 short (1 min) videos describing the project and the methods used can be found at http://steppingupnexus.org.uk/?q=content/stepping-videos (accessed on 5 October 2020). Individual Links are: Innovations https://vimeo.com/333086328/3fa7842282 (accessed on 12 October 2020); Nexus https://vimeo.com/333084383/92cf37e34b (accessed on 12 October 2020); Methods https://vimeo.com/333079298/c74e890daf (accessed on 12 October 2020); Futures https://vimeo.com/333079275/7a3498b734 (accessed on 12 October 2020).

**Author Contributions:** Conceptualization, R.E.F., D.G.; Methodology, P.J.F., N.P.; Software, N.P.; Validation, R.E.F., D.G., P.J.F.; Formal analysis, P.J.F.; Investigation, P.J.F.; Resources, R.E.F.; Data curation, P.J.F.; N.P.; Writing—original draft preparation, P.J.F., N.P.; Writing—review and editing, P.J.F., D.G.; Visualization, N.P.; Supervision, R.E.F.; Project administration, R.E.F.; Funding acquisition, R.E.F. All authors have read and agreed to the published version of the manuscript.

**Funding:** This research was funded by the Engineering and Physical Sciences Research Council grant number EP/N00583X/1.

**Institutional Review Board Statement:** The study was conducted according to the guidelines of the Declaration of Helsinki, and approved by the Ethics Committee of Abertay University, (approval given 2 April 2018).

**Informed Consent Statement:** Informed consent was obtained from all subjects involved in the study.

**Data Availability Statement:** The data presented in this study is available in the Supplementary Materials.

**Acknowledgments:** The authors would like to acknowledge all the other Stepping Up project partners, in particular, Claire Hoolohan, Carly McLachlan, Ian Soutar, James Suckling and Alice Larkin. We would like to thank the many stakeholders who gave us their time during interviews and workshops to enable this research to take place. We would also like to thank the reviewers for improving the paper.

**Conflicts of Interest:** The authors declare no conflict of interest.

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
