# Peer review of "Interactive Visualisation of Sustainability Indicators for Water, Energy and Food Innovations"

_water, doi:10.3390/w13111571_

Round 1
Reviewer 1 Report
Overall, the manuscript is useful and of interest to readers of this journal. In my opinion, the manuscript will be acceptable for publication in water if the authors make revisions that I would classify as minor.
- Abstract: I believe abstract should be condensed version of the full text. Current abstract includes lengthy background information (L10-L19). Overall abstract needs to be rewritten to deliver the purpose and the findings of the research clearly.
- L97, L115, and elsewhere: Please check proper citation style.
- L145: DST is an acronym for what?
- Materials and methods: Differences of study methods between this and previous studies are missing. In L113, a list of decision support tools is provided. What is major difference?
- L213 (Figure 1): Please provide more explanation about figure 1. The meaning of green and white bar (on the top right of the figure) is not clear.
- Conclusion section is missing. I suggest replacement of discussion section into conclusion section because results section already contains a bit of discussion.

Author Response
Reviewer 1
|
Thank-you for your comments, it has helped us to clarify several points that you raised, we confirm each of these below: |
Overall, the manuscript is useful and of interest to readers of this journal. In my opinion, the manuscript will be acceptable for publication in water if the authors make revisions that I would classify as minor.
- Abstract: I believe abstract should be condensed version of the full text. Current abstract includes lengthy background information (L10-L19). Overall abstract needs to be rewritten to deliver the purpose and the findings of the research clearly.
- This has been completely rewritten taking into account the above comment.
- L97, L115, and elsewhere: Please check proper citation style. Amended as requested
- L145: DST is an acronym for what? Decision Support Tool – apologies, now inserted in the text.
- Materials and methods: Differences of study methods between this and previous studies are missing. In L113, a list of decision support tools is provided. What is major difference?
- This comment has been addressed by including a few sentences describing the differences between the tools
- L213 (Figure 1): Please provide more explanation about figure 1. The meaning of green and white bar (on the top right of the figure) is not clear.
- A description of this graphical item has now been included
- Conclusion section is missing. I suggest replacement of discussion section into conclusion section because results section already contains a bit of discussion. Amended as requested.

Reviewer 2 Report
This paper is about developing sustainability indicators that address global environmental challenges holistically rather than individually. The authors frame their task within the so-called Water-Energy-Food (WEF) Nexus, and derive their sustainability indicators from a stakeholder consultation exercise. They make use of three innovative environmental initiatives – anaerobic digestion; insect protein for feed for animals or humans; and the redistribution of surplus food – to illustrate the application of their sustainability indicators.
The paper has an admirable objective – to integrate environmental policy-making - but the way the research is conducted raises four questions in my mind.
First, the elaborate model created by the authors to track or exemplify the complex factors raised by specific environmental innovations does not appear to add much to a non-modelled analysis of these factors. The interactive visualisation tool which is at the heart of the model does not seem to me to expand our understanding of sustainable decision making. Perhaps its main function is simply to give us a visual reminder from a pop-up display figure that environmental decision-making is complex with many factors to be borne in mind?
Second, the authors’ concluding remarks –This visualisation tool, although not providing any specific ‘answers’ should allow for a more playful exploration of the different interlinkages, and to be the seed of conversations around sustainability…we hope that the tool can be used to explore the innovations, the indicators, and the possible future scenarios in a playful way and to be the desired conversation starter that it was intended to be” (lines 338-345) – suggests that their analysis was not designed to produce robust instruments, criteria, metrics or guidelines for sustainable policy but merely suggestive ruminations. This interpretation seems to be confirmed by the following statement on lines 165-168: “By considering these innovations under three different potential future scenarios…we also include some ‘food for thought’, i.e., considerations of how things could be in varying futures, and this should promote debate and discussion around the decision to be made”.
Third, what is the point of the “three different potential future scenarios”? They are set out in Table 1 as
“Share & Connect (Decentralised digital society with high levels of connection between producers, consumers and the environment); Create & Cope (A society troubled by climate change, but with vibrant innovation in services systems catering for most needs); and Big & Smart (A highly centralised society where big infrastructure supplies for basic needs, regulated for transparency and efficiency) (lines 206-207)
In the interactive visualisation link, the three scenarios are described in more detail, but the authors do not explain what we are to make of them and how they are relevant to the argument of the paper. Does it matter which, if any, scenario comes to pass? Should we be trying to nudge our country to one or other scenario?
Fourth, on lines 226-227, the authors say “To allow us to create our Interactive Visualisation Tool we needed to establish which of the SDGs were relevant to each of our selected Sustainability Indicators (SIs)”. But why is it necessary for the sustainability indicators to be linked to the sustainable development goals?
Before this MS is considered for publication, therefore, I would like the authors to address the above four points, and in doing so to explain how their paper adds significant value to our understanding of the problems of environmental decision-making. In short, can they convince us that their visualisation tool is more than a well-designed graphical representation of the truism that environmental policy-making is complex?
Author Response
Reviewer 2
|
Thank-you for your valuable comments, it has helped us to clarify several points that you raised, we confirm each of these below: |
|
|
|
|
Comments and Suggestions for Authors
This paper is about developing sustainability indicators that address global environmental challenges holistically rather than individually. The authors frame their task within the so-called Water-Energy-Food (WEF) Nexus, and derive their sustainability indicators from a stakeholder consultation exercise. They make use of three innovative environmental initiatives – anaerobic digestion; insect protein for feed for animals or humans; and the redistribution of surplus food – to illustrate the application of their sustainability indicators.
The paper has an admirable objective – to integrate environmental policy-making - but the way the research is conducted raises four questions in my mind.
First, the elaborate model created by the authors to track or exemplify the complex factors raised by specific environmental innovations does not appear to add much to a non-modelled analysis of these factors. The interactive visualisation tool which is at the heart of the model does not seem to me to expand our understanding of sustainable decision making. Perhaps its main function is simply to give us a visual reminder from a pop-up display figure that environmental decision-making is complex with many factors to be borne in mind?
- We agree with your comment here, this is the purpose of our visualisation tool ie to provide a focus for discussion , but it also supports decision makers in thinking about possible future scenarios, we feel that we have now clarified this in the text.
Second, the authors’ concluding remarks –This visualisation tool, although not providing any specific ‘answers’ should allow for a more playful exploration of the different interlinkages, and to be the seed of conversations around sustainability…we hope that the tool can be used to explore the innovations, the indicators, and the possible future scenarios in a playful way and to be the desired conversation starter that it was intended to be” (lines 338-345) – suggests that their analysis was not designed to produce robust instruments, criteria, metrics or guidelines for sustainable policy but merely suggestive ruminations. This interpretation seems to be confirmed by the following statement on lines 165-168: “By considering these innovations under three different potential future scenarios…we also include some ‘food for thought’, i.e., considerations of how things could be in varying futures, and this should promote debate and discussion around the decision to be made”.
- We have now clarified this in the text, the tool was designed to promote a discussion point for decision makers.
Third, what is the point of the “three different potential future scenarios”? They are set out in Table 1 as
“Share & Connect (Decentralised digital society with high levels of connection between producers, consumers and the environment); Create & Cope (A society troubled by climate change, but with vibrant innovation in services systems catering for most needs); and Big & Smart (A highly centralised society where big infrastructure supplies for basic needs, regulated for transparency and efficiency) (lines 206-207)
In the interactive visualisation link, the three scenarios are described in more detail, but the authors do not explain what we are to make of them and how they are relevant to the argument of the paper. Does it matter which, if any, scenario comes to pass? Should we be trying to nudge our country to one or other scenario?

Round 2
Reviewer 2 Report
I am pleased to confirm that you have addressed my concerns in your revised version of the paper